# The Allelopathic Potential of *Rosa blanda* Aiton on Selected Wild-Growing Native and Cultivated Plants in Europe

**DOI:** 10.3390/plants10091806

**Published:** 2021-08-30

**Authors:** Katarzyna Możdżeń, Agnieszka Tatoj, Beata Barabasz-Krasny, Anna Sołtys-Lelek, Wojciech Gruszka, Peiman Zandi

**Affiliations:** 1Department of Plant Physiology, Institute of Biology, Pedagogical University of Krakow, Podchorążych 2 St., 30-084 Krakow, Poland; 2Department of Botany, Institute of Biology, Pedagogical University of Krakow, Podchorążych 2 St., 30-084 Krakow, Poland; agnieszka.tatoj@student.up.krakow.pl (A.T.); beata.barabasz-krasny@up.krakow.pl (B.B.-K.); 3Ojców National Park, 32-045 Sułoszowa, Poland; ana_soltys@wp.pl; 4Department of Biological Sciences, Poznań University School of Physical Education, Faculty of Physical Culture in Gorzów Wielkopolski, 66-400 Gorzów Wielkopolski, Poland; elm1@interia.pl; 5International Faculty of Applied Technology, Yibin University, Yibin 644000, China; peiman.zandi@mail.ru

**Keywords:** electrolyte leakage, germination, growth, seedlings, Smooth rose, water content

## Abstract

Invasive plant species are responsible for changing colonized ecosystems by occupying new areas and creating a threat to the functioning of the native flora and fauna populations. Alien plants can produce allelochemicals, substances completely new to indigenous communities. This study investigated the germination seed reactions of *Festuca rubra* L. and *Raphanus sativus* L. var. *radicula* Pers. cv. Rowa on the extracts from the roots, stalks, leaves, and flowers of *Rosa blanda*. Aqueous extracts at concentrations of 1%, 2.5%, and 5% were used in order to determine the allelopathic potential of this alien rose for Europe. With the increase in the concentration of extracts, a decrease in the germination capacity of seeds of the tested species was observed. *R. blanda* extracts inhibited the growth of seedlings. Depending on the concentration and type of the extract, changes in biomass and water content in Red Fescue and Red Radish seedlings were also shown. The highest differences in the electrolyte leakages were noted in seedlings treated with 5% rose extracts. The study showed that the aqueous extracts of *R. blanda* leaves and flowers had the greatest allelopathic potential.

## 1. Introduction

Alien invasive species tend to take over the habitat they enter, transforming its structure and existing vegetation [1]. They compete with other plants for space and access to habitat resources. Their negative impact on the elements of the habitat may be aggravated by secreted allelopathic compounds [2]. When these substances get into the soil, they hinder the growth and development of native species, limiting their occurrence and, thus, biodiversity [3,4,5,6,7]. Allelochemical substances can be present in all parts of plants—roots, rhizomes, stalks, leaves, flowers, fruits, and seeds—and their amount and route of release vary with species [8]. The substances produced by plant secondary metabolism are essential for the interaction of plants with the biotic part of the environment: they help to attract pollinators or seed dispersers, act as a defense against natural enemies and as allelochemicals against potential competitors [9]. Allelopathy is, therefore, a very important plant interaction mechanism. It is a form of communication between plants [10,11]. It plays a key role, both in natural and semi-natural ecosystems and in crops. However, the phenomenon of allelopathy itself is viewed mainly as a type of negative interaction, with only a few exceptions for positive effects [12,13,14].

Among foreign colonizers and potential producers of allelochemical compounds, there may be species of shrubs, often imported from distant corners of the world for decorative purposes [15]. An example is Smooth rose *Rosa blanda* Aiton from the section Cinnamomeae (Figure 1), North American taxon, native to Canada and the United States—from Quebec to Ontario, south to Kansas, and east to Missouri and Ohio [16,17]. Naturally, this species grows in arid non-forest ecosystems such as dry grasslands and prairies. It also tolerates less favorable habitats, e.g., low-nutrient roadsides, especially with sandy soil [16]. Smooth rose has long been cultivated and grown-wild in Europe, including Austria, Finland, France, Lithuania, Germany, Poland, and Hungary [18,19,20,21,22,23]. Its introduction to Europe is estimated in 1773 [24]. For example, in Poland, *R. blanda* currently has the status of a domesticated kenophyte, entering semi-natural communities [15]. However, it is still considered a rare species, identified so far only at 22 stands, scattered mainly in the central and north-western part of the country; these stands were created spontaneously or are a remnant of old, local garden crops [25,26].

Smooth rose was included in the “*List of Species Alien in Europe and to Europe*” [27]. However, its negative effects on native flora components or on other elements of the habitats in which it grows have not been described so far. Nevertheless, direct field observations indicate that this taxon has an invasive potential, enabling it to colonize non-forest habitats. It is resistant to low temperatures, because ground frosts and freezes down to −42.8 °C do not harm it [28]. It is characterized by high reproductive capacity, which is manifested by abundant flowering and fruiting; it is a zoochoric species, mainly ornithochoric. It spreads effectively through the vegetative route through stolons [19,26,29]. Due to the occupied habitats (roadsides, ditches), its expansion could be the most dangerous for species associated with thermophilic grasslands ecosystems. It is also worth bearing in mind that in the case of many species of foreign origin, the lag phase (also known as the waiting or latency phase) has been observed, going on for many years—the period between the appearance of alien species and the disclosure of its invasive interactions [30,31]. That is why it is so important to undertake early research to determine whether alien species may adversely affect native elements of the flora, for example, by secreting allelopathic compounds.

Regarding the chemical composition of all organs, *R. blanda* has not yet been studied [32]. Generally, for the entire family of Rosaceae Juss. the following groups of compounds are given: tannins and their polyphenolic precursors (catechins, leucoanthocyanins), moreover, triterpenes, flavonoids. Phenolic glycosides are also found in this family, for example, phlorizin [33]. In the genus *Rosa* and its related genera, most studies concern the chemical compositions of hypanthium and seeds, both European [34,35,36,37] and North American species [38,39,40,41]. The most frequently mentioned compounds isolated from these structures are: vitamin C, tocopherols, phenols, carotenoids, sugars, and organic and fatty acids [36]. Few studies concern the chemical composition of the petals of North American taxa [42,43]. In the Cinnamomeae section, which includes *R. blanda*, the most common are flavanols (e.g., 3-sophorosides of kaempferol, quercetin) and 11 anthocyanins (e.g., 3-glucosides and 3,5-diglucosides of cyanidin (Cy), pelargonidin and peonidin (Pn), and 3-rutinosides and 3-ρ-coumaroylglucoside-5-glucosides of Cy and Pn) [42,43,44]. The chemical composition of leaves, stalks, and roots has not been studied so far as these organs are not used in medicine and the food industry but may be an important source of allelochemical compounds.

The aim of the experiment was to investigate whether *Rosa blanda* has allelopathic properties that may pose a threat to European native plant species. The experiment analyzed the effect of aqueous extracts of various concentrations from the underground and aboveground organs of *R. blanda* on germination and early seed growth of monocotyledonous and dicotyledonous plants, wild-growing and cultivated. The Petri dishes tests were carried out on seeds of Red Fescue (*Festuca rubra* L.) and Red Radish (*Raphanus sativus* L. var. *radicula* Pers. cv Rowa); the influence of rose extracts on (1) Red Fescue and Red Radish seed germination indexes, (2) elongation growth of seedlings, (3) biomass, and (4) electrolyte leakage from seedling cells was analyzed.

## 2. Results

The germination capacity of *Festuca rubra* seeds expressed as germination percentage (GP) index showed no significant influence of 1% of *R. blanda* organ extracts compared to the control; except 1% leaf extracts had an inhibitory effect on the number of *F. rubra* seeds germinated. With the increase in the concentration of extracts, the seed germination of this species was inhibited compared to the control. The smallest negative impact on the value of GP had extracts from the roots of the studied rose (Figure 2A–D). The coefficient rate of germination (CRG index) was significantly lower than the control for Red Fescue seeds treated with *R. blanda* root and leaf extracts. The other two extracts, from stalks and flowers, had no statistical effect on the value of this parameter (Table 1). Mean germination time (MGT) values were significantly higher for seeds treated with extracts of rose roots and leaves compared to the control. Stalk and flower extracts did not significantly affect MGT values (Table 1).

Each of the rose extracts, regardless of the concentration, had a negative effect on the seed vigor index (SVI) index. The germination index (GI) parameter reached clearly lower values with almost every rose extrac, compared to the control. The exceptions were 1% stalk extracts, which had a positive effect on the values of this index, and 1% flower extracts, which did not significantly affect the GI (Table 1). The allelopathic seeds response (RI) values for Red Fescue seeds were significantly lower than the control, except for seeds treated with 1% extracts of stalks and rose flowers, which significantly increased the RI values (Figure 3A, Appendix A).

The length of roots of *F. rubra* seedlings grown on aqueous extracts from the roots and stalks of *R. blanda* were significantly shorter than in the control. The inhibition of elongation growth was observed with the increase in the concentration of extracts. Contrary to the aboveground parts of seedlings, which were not affected by both root and stalk extracts, in relation to the control seedlings (Figure 4 and Figure 5A,B), the length of whole *F. rubra* seedlings was significantly inhibited by all rose extracts used (in each concentration). Along with the increase in the concentration of allelopathins in the aqueous extracts, the elongation growth of Red Fescue seedlings was lower compared to the seedlings from the control sample (Figure 4 and Figure 5A,B, Appendix A).

Extracts of rose roots, stalks, and flowers had no significant effect on fresh mass of *F. rubra* seedlings (Table 2). Only leaf extracts, in each concentration, had a negative effect on mass in relation to the control. The dry mass of seedlings of this species was significantly higher in those treated with extracts by 1% and 2.5% concentration. An increase in the value of this parameter was observed for seedlings treated with *R. blanda* root and flower extracts. Additionally, the stimulating effect of 5% of stalk extracts on the values of this parameter was demonstrated. In other cases, there was no significant effect of rose extracts on the increase in dry mass. The dry mass/fresh mass (DM/FM) ratio was significantly higher for seedlings treated with all concentrations of root extracts, 5% stalk, 5% leaf extracts, and 2.5% flower extract, compared to the control. The remaining extracts showed no effect on the values of this parameter. Total water content (TWC) was significantly lower in Red Fescue seedlings treated with all types of root extracts and 5% stalk and leaf extracts. The increase in TWC was observed only for seedlings treated with 2.5% flower extract relative to the control (Table 2).

All rose organ extracts increased the electrolyte leakages from Red Fescue seedlings. Compared to the control, only 1% and 5% stalk extracts did not change the destabilization of *F. rubra* cell membranes (Figure 6A).

Aqueous extracts of rose organs showed no significant effect on the germination capacity of *Raphanus sativus* var. *radicula* seeds. A negative effect on the values of GP index was observed for Red Radish seeds treated with 5% root and leaf extracts and 2.5% and 5% of *R. blanda* flowers (Figure 2E–H). The CRG index was unchanged in Red Radish seeds germinating on 1% rose root, stalk, and leaf extracts compared to the control. The remaining rose extracts, in each concentration, caused a statistical decrease in the value of this (Table 1).

Almost all extracts from *R. blanda*, in each concentration, increased the value of the MGT index compared to the control sample. Red Radish seedlings achieved values similar to the control only at 1% root, stalk, and leaf extracts. For the SVI index, the negative effect of all rose extracts was observed.

Only root extracts with concentrations of 1% and 2.5% and 1% rose stalks did not significantly affect the SVI values. The GI index did not change, relative to the control, for seeds with 1% and 2.5% root extracts, all stalk extracts, and 1% rose leaf extracts. In the remaining cases, a decrease in the GI value was demonstrated compared to the control (Table 1). The RI index in each of the extracts, regardless of the concentration and rose organ, was significantly lower than in the control (Figure 3B, Appendix A).

Biometric analysis of the root length of seedlings *R. sativus* var. *radicula* cv. Rowa grown on aqueous extracts of *R. blanda* roots showed significant inhibition of their growth only by 5% of the extracts compared to the control. The length of the aboveground part did not differ between the control and rose root extracts. The shortest radish seedlings were recorded for seeds treated with 5% root extracts compared to the control and other extracts (Figure 5C,D). Rose stalk extracts, at increasing concentrations, inhibited the growth of underground and aboveground radish organs. Significant differences in root length were noted for 2.5% and 5% of extracts compared to the control. In the case of the aboveground parts, only 5% of the stalk extracts inhibited the growth of these organs. Extracts of leaves and flowers, regardless of the concentration, significantly reduced the elongation growth of all organs of radish seedlings as compared to the control seedlings. As the concentration of extracts increased, the elongation growth was smaller and smaller (Figure 5C,D). The length of the whole seedling of *R. sativus* var. *radicula* did not differ from the control in seedlings treated with 1% and 2.5% root extracts and 1% *R. blanda* stalk extracts. In the remaining cases, a significant inhibition of the growth of Red Radish seedlings was demonstrated (Figure 5C,D and Figure 7; Appendix A).

The fresh mass of seedlings of this species was higher, relative to the control, in seedlings treated with 1% extracts of roots, stalks, and rose flowers. Only leaf extracts in each concentration inhibited the growth of fresh mass of seedlings. Root extracts at concentrations of 2.5% and 5% and 2.5% of stalks did not show statistically significant differences in dry mass compared to control seedlings. Additionally, 5% rose stalk extract positively influenced the growth of fresh mass of Red Radish seedlings. Dry mass values did not change compared to the control in seedlings treated with all three concentrations of *R. blanda* root extracts, 1% and 2.5% leaf extracts and 1% flower extracts. In other cases, an increase in the dry mass of seedlings of the tested species was observed. Radish seedling dry to fresh mass ratio was higher than the control in media with 2.5% stalk extracts, all three concentrations of leaf extracts, and 2.5% and 5% from flowers. It was shown that 1% root extracts significantly decreased the value of this parameter compared to the control. No effect on the DM/FM values was found for the remaining two concentrations of root extracts, 5% stalk extracts, and 1% leaf extracts (Table 2). TWC did not change significantly relative to the control values in radish seedlings treated with 2.5% and 5% root extracts, 1% and 5% stalk extracts, and 1% flower extract. The increase in TWC was recorded only in seedlings treated with 1% rose root extracts. The remaining *R. blanda* organ extracts reduced the TWC in radish seedlings compared to the control (Table 2).

The degree of electrolyte leakage (EC) did not differ from the control in the seedlings of *R. sativus* var. *radicula* treated with 1% and 2.5% extracts of rose roots and leaves. The percentage degree of destabilization of cell membranes was higher for seedlings treated with 5% rose root and leaf extracts and all flower extracts. The reduction of the EC value in relation to the control was shown only for radish seedlings grown on media with extracts from the stalks, regardless of their concentration (Figure 6B).

## 3. Discussion

The large group of invasive taxa also includes phanerophytes, i.e., trees or shrubs with woody shoots. North American species are among the most common alien species in Europe and exhibit strong features of invasiveness, among others: *Acer negundo* L., *Padus serotina* (Ehrh.) Borkh., *Quercus rubra* L., *Robinia pseudoacacia* L., and chinese’s of *Ailanthus altissima* (Mill.) Swingle. Among them, there are also Asian roses, among other *Rosa rugosa* Thunb., colonizing the coasts of coastal dunes in many European countries [15,45]. Most of these species found their way to Europe as ornamental plants in parks and gardens and some even in forests or reclaimed land. Animals, especially birds, often contribute to the spread of invasive taxa, for whom their fleshy, tasty fruit is a great attraction [31,46]. In this way, species such as *Padus serotina* and *Rosa rugosa* have increased their area of occurrence. Unfortunately, once they have arrived in new habitats, they quickly became dangerous colonizers and are very difficult to combat [47]. Many invasive species also use allelopathy as an added settlement asset, as already mentioned in the introduction. Allelochemical substances are responsible for modulating the basic and complex physiological mechanisms that occur in plants at various stages of their development [48]. The results of studies on the impact of allelochemical compounds on the increase in the invasiveness of certain plants in ecosystems confirm the important ecological role of this type of interactions [49,50]. For example, in *Padus serotina*, allelopathic properties are already revealed during the youngest stages of development: germination and growth. The compounds secreted by *P. serotina* cause disturbances in colonization of co-occurring seedlings by ectomycorrhizal fungi [51,52]. The negative impact on the soil microflora is one of the methods of eliminating competing individuals [53].

Seed germination in the presence of plant extracts is the starting point for studying the intraspecific and interspecific effects of allelopathy [54]. The allelopathic effect varies with the type of plant organ and the time it decomposes [55]. Seeds subjected to long-term exposure to natural chemicals also become more sensitive to other abiotic and biotic environmental factors [56]. Allelopathic compounds in low concentrations usually have stimulating properties, and in high concentrations, they have a negative effect on seed germination. The impact of compounds on receiver plants may differ between species [57,58,59]. The results of the experiment carried out here with *Rosa blanda* confirm this thesis. Along with the increase in the concentration of extracts, a decrease in the value of germination rates was noted, both for seeds of *Festuca rubra* and *Raphanus sativus* var. *radicula*. Regardless of the rose organ examined, 1% extracts had a positive effect on the seed germination capacity, compared to the control. Extracts from leaves and flowers of *R. blanda* revealed the most inhibitory properties (Figure 2 and Figure 3, Table 1, Table 3 and Table 4). This may be due to the higher concentration of inhibitory compounds present in these parts of the plant compared to the stalks and roots [60]. A similar, inhibitory effect of aqueous flower extracts from *R. damascena* Mill. for germination and growth of seedlings was revealed by Mohammadkhani and Zademobarak [61]. Leaf extracts of *Rosa brunonii* Lindl. (*R. moschata*) inhibited the germination and seedling growth of *Cucumus sativus* L. and *Helianthus annus* L. due to its phytotoxic effect. The extracts had a greater inhibitory effect on cucumber than on sunflower [59]. The reduced germination capacity observed in both cases could be a consequence of a disturbed hormonal balance, especially a change in the activity of gibberellic acid, which regulates the synthesis of de novo amylase during the germination process, as well as disturbances in protein synthesis, activity of enzymes, photosynthesis, and respiration [48]. From an ecological point of view, in highly competitive environments, even slight germination delays can be an advantage for the aggressive neighbor and affect the adaptability of alien taxa. In the long term, this could lead to the extinction of native species and a reduction in local biodiversity [62].

In the conducted study, apart from changes in seed germination, differences were observed in the elongation growth of seedlings growing in the presence of aqueous extracts of *R. blanda* (Figure 4, Figure 5, Figure 6, Table 3 and Table 4). The tested aqueous extracts influenced both the growth of the roots and the aboveground parts of seedlings. The roots of *F. rubra* and *R. sativus* var. *radicula* appeared to be more sensitive to the allelopathic effect than the aboveground parts. The higher sensitivity of the root probably resulted from the time the organ was in contact with the extract. During germination, the seed coat is the first to be exposed to the influence of environmental factors and then the developing roots of the seedling [63]. The reduction in seedling root length can be attributed to restricted cell division due to the presence of allelochemical compounds that can inhibit the action of growth hormones [64]. They can also inactivate respiratory enzymes involved in the oxidative pentose phosphate pathway, necessary for RNA and DNA synthesis, and disrupt the production of ATP and the production of intermediates for the Calvin cycle [65,66]. The consequence of changes in seedling germination and growth in the presence of aqueous extracts from *R. blanda* were differences in biomass production. Depending on the organ and the concentration of allelochemical compounds in the solutions, the extracts had a more or less inhibitory effect, which indicated a different sensitivity of the seedlings. Aqueous extracts of rose roots and stalks caused an increase in seedling mass production in contrast to extracts from leaves and flowers, which inhibited the growth of fresh mass (Table 2). This different allelopathic effect could be due to the presence of different types of allelochemical compounds in the individual extracts [67]. Inhibition of biomass production by extracts from leaves and flowers could result from the activation of enzymes involved in metabolic transformations hindered by allelochemical compounds. Percentage changes in the total water content in seedlings were probably associated with changes in the pH of the environment (Table 5) and the osmotic capacity of seedlings, cell damage, reduced mineral uptake and water by seedling roots [68,69]. The negative impact of rose leaf and flower extracts may be related to the nature of the long-term influence of these organs on plants growing in their surroundings. Allelopathic compounds released by them may have a greater impact on other taxa as a result of their shedding and decay over many years. Allelopathic processes involve the release of chemical compounds that can be toxic into the environment immediately after their production by the plant or they become toxic only after they have been transformed by, i.e., microorganisms [70].

Plants synthesize many chemical compounds, the production of which depends on the existence of precursor molecules and the activation of specialized genes, as well as environmental stimuli and their interactions [71]. Allelochemical substances can destroy the balance between free radical production and the protective system in plant tissue [72]. Responsible for this are, among others, phenolic acids, which after plant death may remain in the extract for weeks or months and accelerate the peroxidation of cell membrane lipids, increasing their permeability [73]. For example, phenol-cinnamic acid, produced by some plants, is a precursor to phenylpropanoids, which cause peroxidation and reduce the H+-ATPase activity of the cell membrane, consequently, reducing the viability of plant roots [74]. In the experiment carried out here, the increase in the degree of destabilization of cell membranes was correlated with the allelopathic effect of aqueous extracts from rose organs (Figure 6). It can be assumed that the high values of electrolyte leakages were the result of a greater number of allelochemical substances decomposing from individual organs of *R. blanda*. Probably, the effect of increasing the destabilization of seedling cell membranes, regardless of the species of the tested seeds, had an impact on the modification of the water–ion balance of cells, the formation and deposition of lignin, and thus, influenced other life processes of developing seedlings. All examined organs of *R. blanda* showed allelopathic properties, although the allelopathic effect was inconclusive in the response of the tested seeds to individual extracts (Figure 2, Figure 3, Figure 4, Figure 5, Figure 6, Figure 7, Table 1, Table 2, Table 3 and Table 4). This type of reaction was most likely caused by the release of various types and amounts of chemicals into the aqueous extracts [10,75,76,77]. The joint action of the chemicals in a mixture is different from that of the individual compounds. In a mixture, the concentration of each chemical can be significantly less than that of a single substance [78]. Their interactions—including antagonism, synergy—usually occur at moderate to high concentrations. At low concentrations they are unlikely or are toxicologically insignificant [79,80]. Various factors simultaneously affect the amount of allelopathins produced. For example, the synergistic interaction between allelopathic stress and thermal stress on ferulic acid content is known [81]. The high content of nitrates or carbon compounds in the soil modifies the allelopathic effect of p-coumaric acid [82]. This suggests that various organic compounds present in soil in non-toxic concentrations may increase the toxicity of allelopathic substances [83]. Different compounds, not necessarily emitted by the same donor plant, can reach the receiving plant at the same time, which can directly or indirectly enhance the activity of a given allelochemical compound [84]. Although the concentrations used in the bioassays discussed here are consistent with those commonly used to assess the phytotoxicity of plant extracts, determining the concentrations of active substances in the field as well as their bioavailability is another and very important step to finally confirm the ecological significance of the presented properties of the rose. The composition of plant secondary metabolites is influenced by age of the plant, the developmental stage of the organs, their location (i.e., in the case of leaves), and the method of preparation of the extracts. In this experiment, extracts have been prepared from separate plant material, not from a stock solution, which might also have influenced the results.

The movement of allelopathic compounds in soil involves complex processes and generally occurs through mass flow, diffusion, and trapping by plant roots. During this movement, allelochemicals undergo retention, transformation, and some degradation processes [85].

Thus, identifying factors and plants that may interfere with the germination and growth of other plants is one of the most difficult ecosystem management strategies [86,87]. Advances in this type of research will allow us to understand the phenomenon of allelopathy in the struggle of native species against alien species [10,88]. Invaded ecosystems will almost certainly undergo irreversible changes; however, the evolution of resistance by natives may provide some degree of ‘biotic resistance’ and ultimately re-establish some degree of community equilibrium and species coexistence [62]. In Europe, *R. blanda* can be considered a potentially invasive species [26]. Its allelopathic properties additionally confirm this potential. It is also worth mentioning that this species can hybridize with highly invasive *R. rugosa*. Study has revealed frequent bidirectional hybridization and rare introgression in populations of these species. The repeated presence of hybrids indicated weakness in early reproductive barriers. This type of hybridization could eventually lead to the genetic assimilation of *R. blanda* in mixed populations and the emergence of invasive hybrid genotypes, a phenomenon that is of environmental concern due to the growing number of alien species worldwide [89].

## 4. Materials and Methods

### 4.1. Plant Material

*Rosa blanda* specimens were obtained in June 2020 from a site in north-west Poland, on the side of the road between Strzelce Krajeński and Wielisławice (52°53′31.75″ N; 15°30′8.12″ E). The rose organs, harvested at the same time, were in full vegetative development. The collected plants were selected in terms of damage and organ health (underground and aboveground). Uninfected and undamaged plants, morphologically similar, were selected for the experiments. The plants, divided into parts (roots, stalks, leaves, and flowers), were dried in the laboratory in the dark at room temperature 23 ± 2 °C with an average air humidity of 60–70%. The drying time of individual rose organs was varied due to their anatomical structure. The flower petals and leaves were dried for 7 days, and the roots and stalks for 10 to 12 days. The dried material was stored in paper bags for the duration of the experiment in the dark at room temperature. Red Radish seeds (*Raphanus sativus* var. *radicula* cv. Rowa) bought from Polan sp.z.o.o. Breeding and Seed Horticulture in Kraków (Poland). Red Fescue seeds (*Festuca rubra*) were bought from DLF SeeDs (Hladké Životice, Czech Republic).

### 4.2. Extract Preparation

The dried parts of *R. blanda* (roots, stalks, leaves, and flowers) were each separately ground in a mortar and aqueous extracts at various percentage concentrations (1%, 2.5%, and 5%) were prepared from them [90]. For example, a 1% root extract was made by preparing 1 g of dried rose root, which was flooded over with 99 mL of distilled water; respectively—extract 2.5%—2.5 g dry material + 97.5 mL distilled water; 5% extract—5 g dry material + 95 mL distilled water. In the same way, extracts from all rose organs (parts) were prepared. For the extraction of chemical substances, the plant material flooded with water was stored for 24 h in the dark at a room temperature of 23 ± 2 °C (samples were protected with cellophane). After this time, the aqueous extracts were filtered through gauze and stored in a refrigerator at 8 ± 2 °C throughout the experiment.

### 4.3. PH of Extracts

The pH of each extract prepared from individual parts of *R. blanda* was measured by immersion in the pH electrode solution (Elmetron, Zabrze, Poland). The mean pH values of the five replicates were determined for each extract (Table 5).

As the concentration of the extracts increased, the pH values of the aqueous extracts from *R. blanda* organs were higher. Only in the case of the root was the opposite observed; the pH of this solution was the highest in 1% extract and the lowest in 2.5%.

### 4.4. Seeds Preparation and Germination Conditions

Red Fescue and Red Radish seeds (each separately) were sterilized in 1% acetone solution for 1 min, then rinsed three times with distilled water. Twenty-five seeds were placed in sterile Petri dishes (Ø 9 cm) with three layers of filter paper, moistened with an appropriate rose extract (5 mL every other day). The control group consisted of seeds placed on Petri dishes treated only with distilled water. The seeds were placed in the Petri dishes in the dark, at room temperature 23 ± 2 °C, a relative humidity of about 60–70%. The number of germinated seeds was checked every 24 h for 7 days. The experiment was performed in three replicates (*n* = 10) for each concentration and type of rose extract and controls.

### 4.5. Germination Indexes

The germination capacity of Red Fescue and Red Radish seeds was performed by the germination indexes. GP—germination percentage (global method), GI—germination index [90], MGT—mean germination time [5], SVI—seedling vigor index for cumulative seeds [91], CRG—coefficient rate of germination [92], MGT—mean germination time [93], RI—allelopathic seeds response [94].
GP (%) = [number of seeds germinated each day/total number of seeds used in the bioassay] × 100(1)
MGT (day) = ƩDn/ƩnD—the number of days counted from the beginning of germination, N—number of seeds that appeared on day D(2)
GI (%) = [Number of germinated seeds/Days of first count] + [Number of germinated seeds/Days of second count] + … + [Number of germinated seeds/Days of last or final count](3)
CRG (%) = [(n3 + n7)/((n3 × T3) + (n7 × T7))] × 100n3 and n7 = number of germinated seeds on time T3 and T7, T3 and T7 = 3 and 7 days after germination(4)
SVI (a.u.) = (Seedling length (cm) × Germination percent)/100(5)
RI (a.u.) = T/C − 1 (T < C)C—control germination speed, T—treatment germination speed(6)

### 4.6. Seedling Elongation Growth

Biometric analysis of Red fescue and Red radish seedlings grown on various extracts from *R. blanda* organs were based on the analysis of the length of the underground and aboveground parts of the seedlings using the traditional method, using a caliper (Topex 31C615, Kraków, Poland) with an accuracy of 0.1 mm. The germination inhibition index (IP) was also determined using the formula of Mominul Islam et al. [95].
IP (%) = (1 − (LE/LC)) × 100LE—seedling length (mm) treated with the aqueous extract, LC—seedling length (mm) treated with the distilled water (control group)(7)

### 4.7. Seedlings Biomass

After 7 days of germination of Red Fescue and Red Radish seeds, fresh and dry mass (g) was determined for single seedlings on a scale (Ohaus Adventurer Pro, Morris County, NJ, USA), with an accuracy of 0.0001 g. Dry mass of seedlings was determined after 48 h of drying at 105 °C in a dryer (WAMED SUP 100, Zabrze, Poland).

Based on the obtained results, the total water content was determined [96].
WC (%) = 100 − [(DM × 100)/FM]WC—water content, DM—dry mass, FM—fresh mass(8)

### 4.8. Electrolyte Leakage

The percentage of electrolyte leakage, reflecting the degree of destabilization of cell membranes of Red Fescue and Red Radish seedlings, was measured according to the method of Szafraniec et al. [97]. Single seedlings of Red Fescue or Red Radish were placed in 30 mL of distilled water in polypropylene vials and shaken for 3 h on a shaker (Labnet, Rocker, Edison, NJ, USA). After this time, each of the samples was additionally vortexed for 10 s. Using a CX-701 conductometer with an electrode (K = 1.02) (Elmetron, Zabrze, Poland), the leakage of electrolytes from the living cells was measured (E1). Then, the seedlings in vials with water were frozen for 24 h at temperature −25 °C in order to macerate the cells. After this time, the seedling samples were thawed and subjected to the same shaking and measurement procedure as the viable seedling samples to determine total electrolyte leakage (E2). From the obtained results, the percentage of electrolyte leakage was determined according to the formula:EL (%) = (E1/E2) × 100EL—electrolyte leakage, E1—EL from live seedlings, E2—EL from dead seedlings(9)

### 4.9. Statistic Analysis

The experiment was carried out in three replicates on 25 Red Fescue seeds and 25 Red Radish seeds separately for each type and concentration of aqueous extracts from *R. blanda* organs and a control sample (distilled water). The obtained mean results (*n* = 10, ±SD) were subjected to one-way ANOVA statistical analysis. Differences between extracts for each type of extract and seed species are marked with different letters in the tables, both in columns and in rows, using Duncan’s test at *p* ≤ 0.05 in the program StatSoft, Inc. Poland, 2018. STATISTICA (data analysis software system), version 13.1.

## 5. Conclusions

(1) With increasing concentrations of *Rosa blanda* Aiton extracts. their inhibitory effect on the capacity and speed of germination of Red Fescue and Red Radish seeds was found; root extracts had the lowest effect on the values of germination rates, and rose leaf and flower extracts had the highest effect. (2) Similar negative reactions of seedlings to rose extracts were observed for the parameters of elongation growth; early contact of germinating seeds with allelopathic substances affected the roots to a higher extent than the aboveground parts of the seedlings. (3) Compared to the control, the fresh mass of Red Fescue and Red Radish seedlings was greater in seedlings grown on root and stalk extracts and lower in those treated with extracts of rose leaves and flowers; dry mass significantly differed from control values in all tested cases; the percentage of water content in the seedlings differed from the control depending on the concentration and type of extract. (4) Aqueous extracts of *R. blanda* organs destabilized the cell membranes of Red Fescue and Red Radish seedlings—with the increase in the concentration of extracts, an increase in the electrolyte leakages was observed, regardless of the type of extract. Compared to the control, the 1% extracts had the lowest impact on membrane structures and 5% the highest.

Susceptibility of seedlings of *Festuca rubra* L. and *Raphanus sativus* L. var*. radicula* Pers. cv. Rowa most likely resulted from the presence of allelochemical substances in the prepared aqueous extracts from *R. blanda* organs. From the tests carried out in laboratory conditions, it can be concluded that *R. blanda* has an allelopathic potential that can threaten both wild and commonly cultivated species in Europe. Therefore, the occurrence of this species should be monitored. As a potentially invasive taxon, it should be eliminated from natural and semi-natural habitats.

## Figures and Tables

**Figure 1 plants-10-01806-f001:**
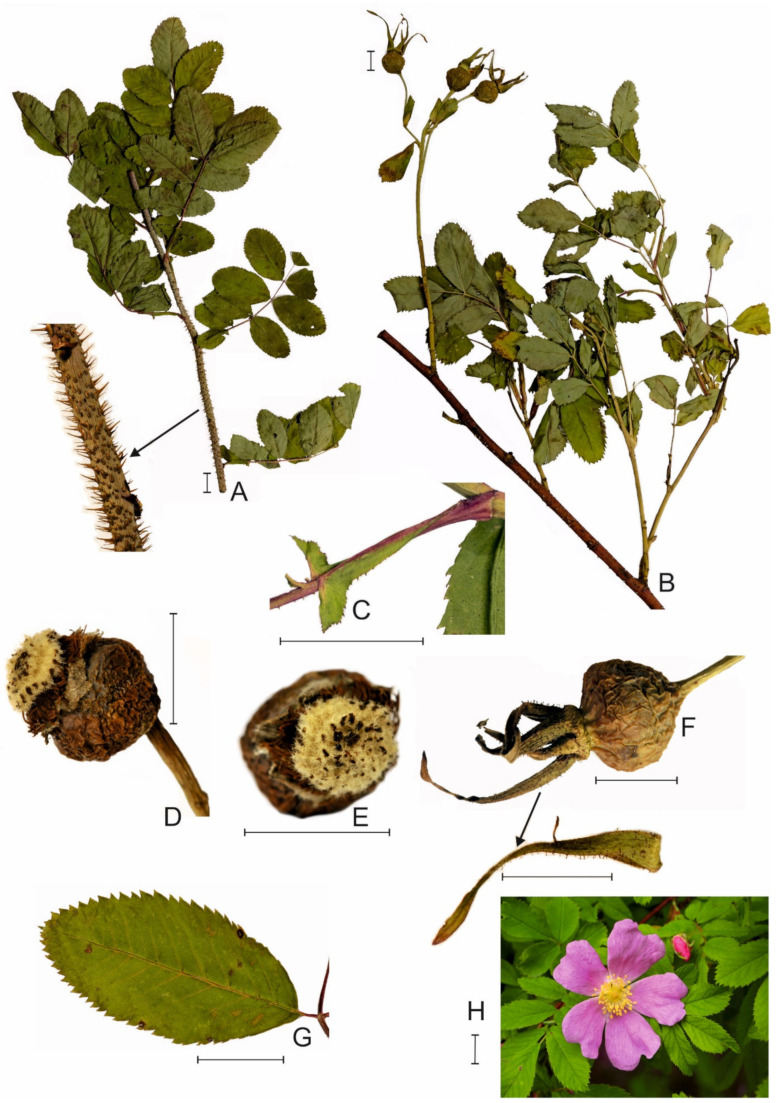
Characteristic features of *Rosa blanda* Aiton; (**A**)—part of long shoot, (**B**)—part of fruiting short shoot, (**C**)—stipule, (**D**,**E**)—fruit, (**F**)—fruit with glandular sepal, (**G**)—leaflet, (**H**)—flower; solid bar = 1 cm (Photo Anna Sołtys-Lelek; specimen from Poland, collected by Wojciech Gruszka between Strzelce Krajeńskie and Wielisławice, 2017).

**Figure 2 plants-10-01806-f002:**
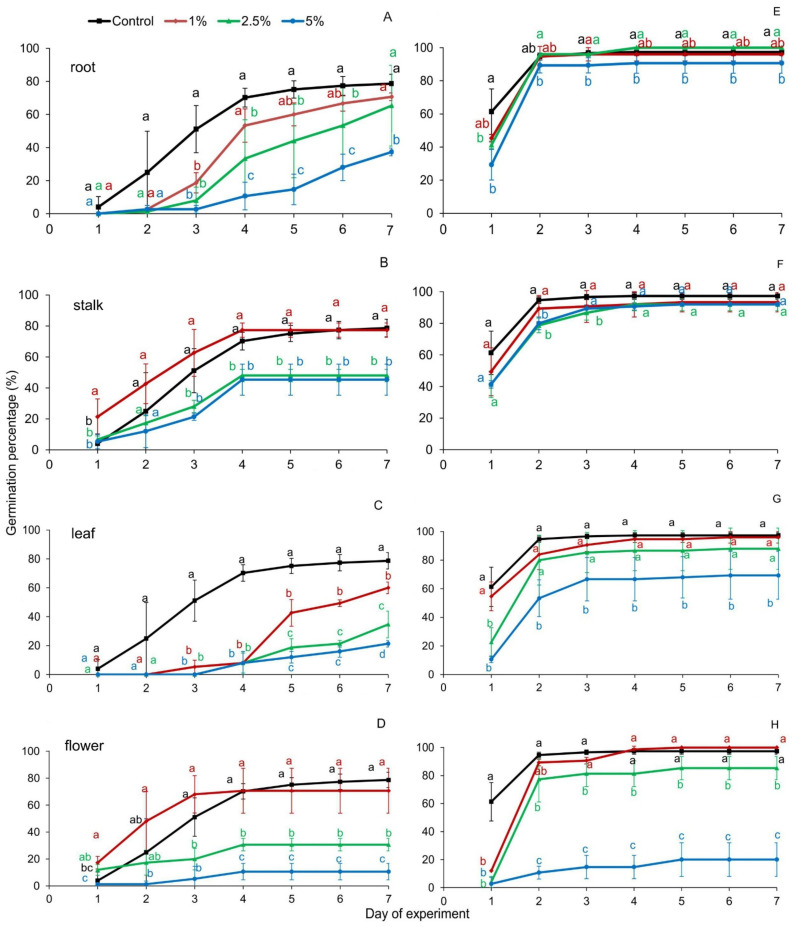
Germination percentage of *Festuca rubra* L. (**A**–**D**) and *Raphanus sativus* L. var. *radicula* Pers. cv. Rowa (**E**–**H**) seeds, treated with organ extracts of *Rosa blanda* Aiton with various concentrations (1%, 2.5%, 5%); aqueous extracts from the following parts of the rose: (**A**,**E**)—root, (**B**,**F**)—stalk, (**C**,**G**)—leaf, (**D**,**H**)—flower; the mean values of the three replicates (*n* = 10, ±SD) marked with different letters are significantly different according to Duncan’s test *p* ≤ 0.05.

**Figure 3 plants-10-01806-f003:**
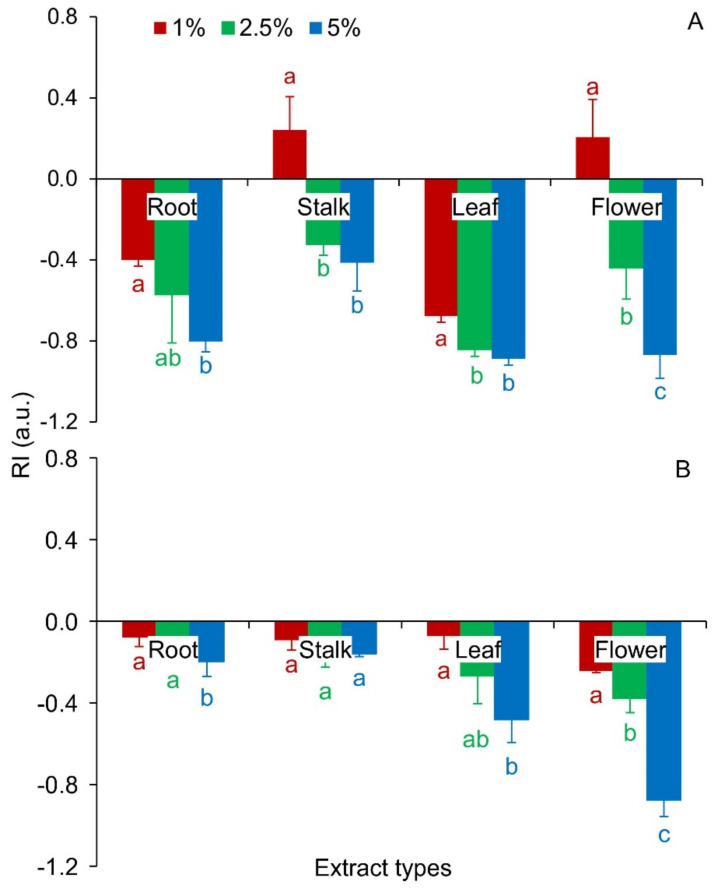
RI index—allelopathic seeds response for *Festuca rubra* L. (**A**) and *Raphanus sativus* L. var. *radicula* Pers. cv. Rowa (**B**) seeds treated with aqueous extracts from organs of *Rosa blanda* Aiton, with various concentrations (1%, 2.5%, 5%); the mean values of the three replicates (*n* = 10, ±SD) marked with different letters are significantly different according to Duncan’s test *p* ≤ 0.05.

**Figure 4 plants-10-01806-f004:**
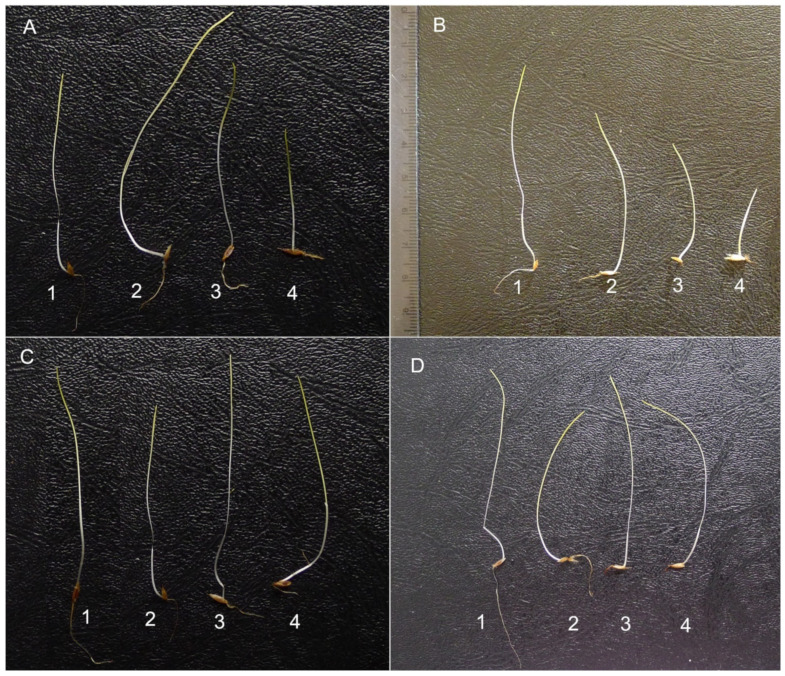
Seedlings of *Festuca rubra* L. grown on aqueous extracts from the organs of *Rosa blanda* Aiton at various percentage concentrations (1%, 2.5%, and 5%); organs: (**A**)—flower, (**B**)—leaf, (**C**)—stalk, (**D**)—root; 1—control (distilled water); the influence of the extracts was analyzed separately at three weekly intervals, hence, the different control view, 2—1% extract, 3—2.5% extract, 4—5% extract (Photo Beata Barabasz-Krasny).

**Figure 5 plants-10-01806-f005:**
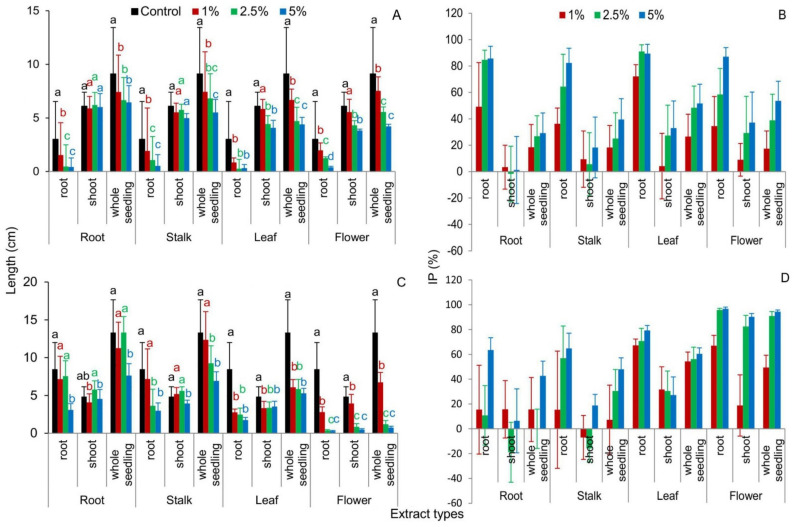
The length of seedlings expressed in (cm) and in (% of control—IP index) of *Festuca rubra* L. (**A**,**B**) and *Raphanus sativus* L. var. *radicula* Pers. cv. Rowa (**C**,**D**) treated with aqueous extracts from the organs of *Rosa blanda* Aiton with various concentrations (1%, 2.5%, 5%); the mean values of the three replicates (*n* = 10, ±SD) marked with different letters are significantly different according to Duncan’s test *p* ≤ 0.05.

**Figure 6 plants-10-01806-f006:**
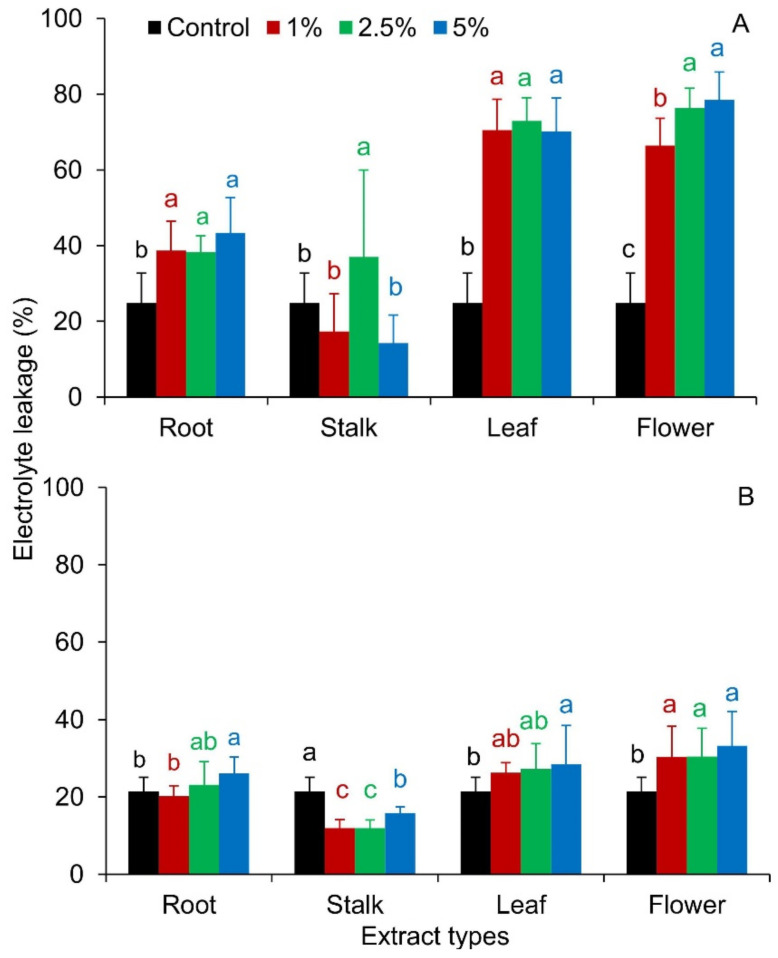
Electrolyte leakage from seedlings of *Festuca rubra* L. (**A**) and *Raphanus sativus* L. var. *radicula* Pers. cv. Rowa (**B**) treated with aqueous extracts from the organs of *Rosa blanda* Aiton with various concentrations (1%, 2.5%, 5%); the mean values of the three replicates (*n* = 10, ±SD) marked with different letters are significantly different according to Duncan’s test *p* ≤ 0.05.

**Figure 7 plants-10-01806-f007:**
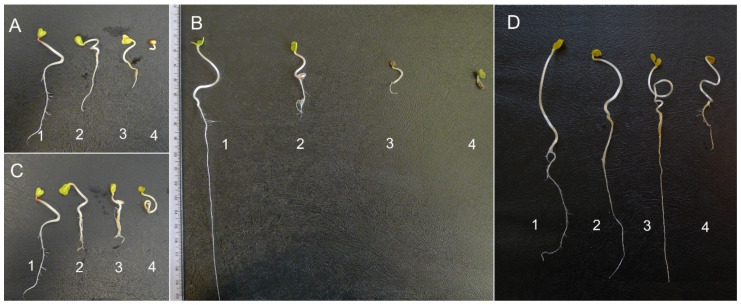
Seedlings of *Raphanus sativus* L. var. *radicula* Pers. cv. Rowa grown on aqueous extracts from the organs of *Rosa blanda* Aiton at various percentage concentrations (1%, 2.5%, and 5%); organs: (**A**)—flower, (**B**)—leaf, (**C**)—stalk, (**D**)—root; 1—control (distilled water); the influence of the extracts was analyzed separately at three weekly intervals, hence, the different control view, 2—1% extract, 3—2.5% extract, 4—5% extract (Photo Katarzyna Możdżeń).

**Table 1 plants-10-01806-t001:** Seed germination rates of *Festuca rubra* L. (A) and *Raphanus sativus* L. var. *radicula* Pers. cv. Rowa (B) seeds, treated with aqueous extracts from the organs of *Rosa blanda* Aiton with various concentrations (1%, 2.5%, 5%).

Extract Types (%)	CRG (%)		MGT (Day)		SVI (a.u.)		GI (%)	
	A	B	A	B	A	B	A	B
	Root							
Control	20.22 a ±1.24	23.94 a ±0.43	4.96 c ±0.30	4.18 b ±0.07	71.6 a ±15.61	129.7 a ±42.17	22.55 a ±6.26	53.70 a ±3.17
1	18.51 b ±0.25	23.53 ab ±0.05	5.40 b ±0.07	4.25 ab ±0.01	52.4 b ±11.07	108.0 a ±33.05	13.52 b ±0.68	49.40 ab ±2.35
2.5	17.55 b ±0.61	23.27 b ±0.07	5.70 ab ±0.20	4.30 a ±0.01	43.5 b ±9.18	133.2 a ±21.09	9.67 b ±5.38	49.32 ab ±0.58
5	17.13 b ±0.67	23.09 b ±0.16	5.84 a ±0.22	4.33 a ±0.03	24.1 c ±5.20	69.2 b ±14.21	4.46 c ±1.15	43.16 b ±4.00
F	10.10	3.80	20.87	5.22	36.88	8.64	8.32	8.45
*p*	0.01 *	0.04 *	0.000 *	0.02 *	0.000 *	0.000 *	0.002 *	0.003 *
	**Stalk**							
Control	20.22 a ±1.24	23.94 a ±0.43	4.96 a ±0.30	4.18 b ±0.07	71.6 a ±15.61	129.7 a ±42.17	22.55 b ±6.26	53.70 a ±3.17
1	21.80 a ±0.81	23.61 a ±0.51	4.59 a ±0.17	4.24 b ±0.09	57.5 b ±11.75	115.4 a ±34.79	30.57 a ±6.00	48.69 a ±2.55
2.5	20.65 a ±0.19	23.15 b ±0.33	4.84 a ±0.05	4.32 a ±0.06	32.7 bc ±8.50	85.2 b ±21.18	15.28 c ±1.24	44.86 a ±3.26
5	20.23 a ±1.22	23.20 b ±0.12	4.96 a ±0.30	4.31 a ±0.02	25.0 c ±6.50	63.7 b ±11.23	13.22 c ±3.14	45.16 a ±0.75
F	2.63	3.54	2.07	4.40	42.25	10.89	5.94	7.07
*p*	0.09	0.03 *	0.15	0.03 *	0.000 *	0.000 *	0.008 *	0.006 *
	**Leaf**							
Control	20.22 a ±1.24	23.94 a ±0.43	4.96 b ±0.30	4.18 c ±0.07	71.6 a ±15.61	129.7 a ±42.17	22.55 a ±6.26	53.70 a ±3.17
1	16.91 b ±0.53	23.53 ab ±0.46	5.92 a ±0.19	4.25 bc ±0.08	40.1 b ±9.19	58.5 b ±9.70	7.28 b ±0.69	49.80 a ±3.40
2.5	16.66 b ±0.91	22.75 bc ±0.70	6.02 a ±0.34	4.40 ab ±0.14	16.3 c ±5.19	51.4 b ±11.29	3.56 c ±0.77	39.34 b ±7.36
5	16.90 b ±0.83	22.15 c ±0.23	5.93 a ±0.29	4.52 a ±0.05	9.4 c ±2.83	36.6 b ±4.55	2.53 c ±0.72	27.82 c ±6.03
F	12.27	6.41	12.27	7.82	93.64	34.92	16.38	23.59
*p*	0.000 *	0.009 *	0.000 *	0.004 *	0.000 *	0.000 *	0.000 *	0.000 *
	**Flower**							
Control	20.22 ab ±1.24	23.94 a ±0.43	4.96 ab ±0.30	4.18 c ±0.07	71.6 a ±15.61	129.7 a ±42.17	22.55 b ±6.26	53.70 a ±3.17
1	22.23 a ±0.38	22.23 b ±0.07	4.50 b ±0.08	4.50 b ±0.01	53.1 b ±8.62	67.3 b ±13.14	29.42 ab ±6.79	40.63 b ±0.46
2.5	21.92 a ±0.75	22.08 b ±0.14	4.57 b ±0.15	4.53 ab ±0.03	17.1 c ±5.51	10.3 c ±4.01	12.66 c ±3.48	33.40 c ±3.77
5	19.24 b ±1.83	21.22 b ±1.13	5.23 a ±0.47	4. 72 a ±0.25	4.5 d ±1.44	1.5 c ±0.37	2.21 d ±3.19	6.69 d ±4.36
F	5.95	14.83	5.95	11.45	106.9	65.04	12.44	109.3
*p*	0.008 *	0.000 *	0.008 *	0.001 *	0.00 *	0.000 *	0.000 *	0.000 *

CRG—coefficient rate of germination, MGT—mean germination time, SVI—seed vigor index, GI—germination index; mean values from three replicates (*n* = 10, ± SD) marked with different letters differ significantly (in columns, separately for a given species of seed and type of extract) according to Duncan’s test * *p* ≤ 0.05.

**Table 2 plants-10-01806-t002:** Fresh and dry mass and total water content in seedlings of *Festuca rubra* L. (A) and *Raphanus sativus* L. var. *radicula* Pers. cv. Rowa (B) treated with aqueous extracts from the organs of *Rosa blanda* Aiton with various concentrations (1%, 2.5%, 5%).

Extract Type (%)	Fresh Mass (g)		Dry Mass (g)		Dry Mass/Fresh Mass (a.u.)		Total Water Content (%)	
	A	B	A	B	A	B	A	B
	Root							
Control	0.0075 a ±0.001	0.0874 b ±0.020	0.0003 b ±0.0003	0.0049 a ±0.001	0.0443 b ±0.046	0.0560 a ±0.009	95.57 a ±4.61	94.40 b ±0.87
1	0.0072 a ±0.002	0.1219 a ±0.032	0.0008 a ±0.0005	0.0047 a ±0.001	0.1124 a ±0.073	0.0396 b ±0.009	88.76 b ±7.29	96.04 a ±0.95
2.5	0.0074 a ±0.002	0.1145 ab ±0.032	0.0008 a ±0.0004	0.0058 a ±0.001	0.1139 a ±0.065	0.0519 a ±0.010	88.61 b ±6.50	94.81 b ±1.01
5	0.0062 a ±0.001	0.1027 ab ±0.030	0.0006 ab ±0.0003	0.0051 a ±0.002	0.0994 a ±0.046	0.0489 a ±0.009	90.06 b ±4.60	95.11 b ±0.95
F	1.74	2.65	3.68	0.94	3.12	5.10	3.12	5.10
*p*	0.18	0.06	0.021 *	0.43	0.038 *	0.005 *	0.038 *	0.005 *
	**Stalk**							
Control	0.0075 a ±0.001	0.0874 c ±0.020	0.00003 b ±0.0003	0.0049 c ±0.001	0.0443 b ±0.046	0.0560 bc ±0.009	95.57 a ±4.61	94.40 ab ±0.87
1	0.0074 a ±0.001	0.1280 a ±0.037	0.0005 ab ±0.003	0.0061 bc ±0.001	0.0692 ab ±0.044	0.0490 c ±0.009	93.08 ab ±4.40	95.10 a ±0.93
2.5	0.0072 a ±0.002	0.1011 bc ±0.018	0.0005 ab ±0.0004	0.0067 ab ±0.001	0.0747 ab ±0.056	0.0667 a ±0.013	92.53 ab ±5.59	93.32 c ±1.27
5	0.0074 a ±0.002	0.1200 ab ±0.031	0.0008 a ±0.0003	0.0075 a ±0.001	0.0980 a ±0.039	0.0640 ab ±0.011	90.20 b ±3.91	93.60 bc ±1.12
F	0.14	3.84	2.70	6.04	2.23	5.05	2.23	5.05
*p*	0.94	0.018 *	0.01 *	0.002 *	0.02 *	0.005 *	0.02 *	0.005 *
	**Leaf**							
Control	0.0075 a ±0.001	0.0874 a ±0.020	0.0003 a ±0.0003	0.0049 b ±0.001	0.0443 b ±0.046	0.0560 b ±0.009	95.57 a ±4.61	94.40 a ±0.87
1	0.0065 b ±0.001	0.0467 b ±0.013	0.0006 a ±0.0003	0.0058 ab ±0.002	0.0839 b ±0.051	0.1239 a ±0.017	91.61 a ±5.08	87.61 b ±1.74
2.5	0.0064 b ±0.001	0.0530 b ±0.008	0.0005 a ±0.0004	0.0061 ab ±0.001	0.0732 b ±0.070	0.1175 a ±0.031	92.68 a ±7.01	88.25 b ±3.11
5	0.0037 c ±0.001	0.0561 b ±0.011	0.0006 a ±0.0004	0.0063 a ±0.001	0.1669 a ±0.099	0.1156 a ±0.031	83.31 b ±9.94	88.44 b ±3.09
F	2.27	1.28	1.26	1.92	5.68	1.58	5.68	1.58
*p*	0.000 *	0.000 *	0.32	0.15	0.003 *	0.000 *	0.003 *	0.000 *
	**Flower**							
Control	0.0075 a ±0.001	0.0874 b ±0.020	0.0003 b ±0.0003	0.0049 b ±0.001	0.0443 b ±0.046	0.0560 c ±0.009	95.57 a ±4.61	94.40 a ±0.87
1	0.0071 a ±0.001	0.1041 a ±0.015	0.0005 ab ±0.0003	0.0063 b ±0.002	0.0746 ab ±0.037	0.0617 c ±0.018	92.54 ab ±3.71	93.83 a ±1.77
2.5	0.0078 a ±0.001	0.0670 c ±0.010	0.0007 a ±0.0002	0.0094 a ±0.002	0.0875 a ±0.030	0.1434 b ±0.037	91.25 b ±2.95	85.66 b ±3.66
5	0.0070 a ±0.002	0.0294 d ±0.011	0.0004 b ±0.0003	0.0085 a ±0.003	0.0579 ab ±0.040	0.2980 a ±0.071	94.21 a ±4.03	70.20 c ±7.06
F	0.65	4.92	3.01	9.63	2.38	6.36	2.38	6.36
*p*	0.59	0.000 *	0.04 *	0.000 *	0.03 *	0.000 *	0.03 *	0.000 *

Three replicates, mean values (*n* = 10, ± SD) marked with different letters are significantly different according to Duncan’s test * *p* ≤ 0.05.

**Table 3 plants-10-01806-t003:** Comparison of the studied germination and growth parameters of *Festuca rubra* L. seedlings grown on the aqueous extracts from the organs of *Rosa blanda* Aiton with different percentage concentrations (1%, 2.5%, 5%).

	Extracts (%)	Root			Stalk			Leaf			Flower		
Parameters		1	2.5	5	1	2.5	5	1	2.5	5	1	2.5	5
GP													
CRG													
MGT													
SVI													
GI													
RI													
Length (whole seedling)													
FM													
DM													
DM/FM													
TWC													
EC													

No differences relative to the control—blue, positive effect relative to the control—green, negative effect relative to the control—red; GP—germination percentage, CRG—coefficient rate of germination, MGT—mean germination time, SVI—seed vigor index, GI—germination index, RI—allelopathic seeds response, FM—fresh mass, DM—dry mass, TWC—total water content, EC—electrolyte leakage.

**Table 4 plants-10-01806-t004:** Comparison of the studied germination and growth parameters of *Raphanus sativus* L. var. *radicula* Pers. cv. Rowa seedlings grown on the aqueous extracts from the organs of *Rosa blanda* Aiton with different percentage concentrations (1%, 2.5%, 5%).

	Extracts (%)	Root			Stalk			Leaf			Flower		
Parameters		1	2.5	5	1	2.5	5	1	2.5	5	1	2.5	5
GP													
CRG													
MGT													
SVI													
GI													
RI													
Length (whole seedling)													
Fresh mass													
Dry mass													
DM/FM													
TWC													
EC													

No differences relative to the control—blue, positive effect relative to the control—green, negative effect relative to the control—red; GP—germination percentage, CRG—coefficient rate of germination, MGT—mean germination time, SVI—seed vigor index, GI—germination index, RI—allelopathic seeds response, FM—fresh mass, DM—dry mass, TWC—total water content, EC—electrolyte leakage.

**Table 5 plants-10-01806-t005:** PH values of aqueous organ extracts *Rosa blanda* Aiton; the mean values of the five replicates (±SD) marked with different letters differ significantly (in column) according to Duncan’s test *p* ≤ 0.05.

	Parts of Rosa	Root	Stalk	Leaves	Flowers
Extracts [%]					
1		4.99 a ±0.02	4.56 b ±0.01	4.12 c ±0.03	4.85 b ±0.01
2.5		4.12 c ±0.01	4.47 c ±0.02	4.55 b ±0.01	4.86 b ±0.01
5		4.55 b ±0.01	4.62 a ±0.01	4.86 a ±0.07	5.04 a ±0.01

## Data Availability

Not applicable.

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
