# Peer review of "The Allelopathic Potential of Rosa blanda Aiton on Selected Wild-Growing Native and Cultivated Plants in Europe"

_plants, 2021, doi:10.3390/plants10091806_

Round 1

Reviewer 1 Report

Comments:
line 103: instead of: "Smooth Rose Aiton Soft Rose" more correctly would be Smooth Rose.

line 107 should be explained here (at first appearance) what “GP index” means. Identical to line 112 for “CRG index”, l. 115 for “MGT values”; l. 119 for “SVI index” and “GI”; l. 122 for “RI”; 1. 141 "DM / FM"; 1. 146 "TWC";
l. 150 probably wrong "percentage nasion".

Author Response

Reviewer 1,

We would like to thank you for review this manuscript. All suggestion have been added in the text.

Open Review

(x) I would not like to sign my review report

( ) I would like to sign my review report

English language and style

( ) Extensive editing of English language and style required

( ) Moderate English changes required

( ) English language and style are fine/minor spell check required

(x) I don't feel qualified to judge about the English language and style

Yes

Can be improved

Must be improved

Not applicable

Does the introduction provide sufficient background and include all relevant references?

(x)

( )

( )

( )

Is the research design appropriate?

(x)

( )

( )

( )

Are the methods adequately described?

(x)

( )

( )

( )

Are the results clearly presented?

(x)

( )

( )

( )

Are the conclusions supported by the results?

(x)

( )

( )

( )

Comments and Suggestions for Authors

Comments:

Reviewer: line 103: instead of: "Smooth Rose Aiton Soft Rose" more correctly would be Smooth Rose.

Authors: It has been corrected. It has been deleted Smooth Rose.

Reviewer: line 107 should be explained here (at first appearance) what “GP index” means. Identical to line 112 for “CRG index”, l. 115 for “MGT values”; l. 119 for “SVI index” and “GI”; l. 122 for “RI”; 1. 141 "DM / FM"; 1. 146 "TWC";
l. 150 probably wrong "percentage nasion".

Authors: Full names of germination indexes have been added according to Reviewer 1 suggestion. Percentage nasion has been corrected in English language.

Sincerely Yours,

Authors

Reviewer 2 Report

The manuscript “The allelopathic potential of Rosa blanda Aiton into the native wild-growing and cultivated plants in Europe” addresses one of the crucial issues threatening biodiversity globally. The authors did well by identifying the problems and using several indices to estimate the extent of threats posed by this invasive plant. However, few things need to be addressed before the manuscript may be suitable for publication.

General comments

  1. The authors collected different organs and prepared several concentrations of extracts. For instance, they prepared 1% concentration using 1 g of dried plant material and 99 ml of water, 2.5% concentration by 2.5 g of plant material and 98.5 ml of water, and so on. The composition of plant secondary metabolites could be influenced by the plant’s age, developmental stage of organs and even their positions (in the case of leaves). This could impact the solutions prepared rather than the effects of concentrations reported in the manuscript. I recommend the authors repeat the experiment by first preparing a stock solution or a stock crude extract from the respective plant organs (homogenised) and diluting to their desired concentrations. Check whether this could reproduce the results.
  2. The authors did not measure or mention the extract volume used to ‘moisten filter papers”. Was the volume equal for all treatments? If not, then you would expect some filter papers to dry quick, impacting seedling germination.
  3. The manuscript needs some polishing. I have done a few editing.
  4. In the results section, authors should include the test statistics (F-values) and P-values.

Minor comments

  1. Line 18: the word “normal” could be deleted.
  2. Line 20: “In the study investigated” should be changed to “In the study investigated we investigated” or “this study investigated”.
  3. Line 24: “were” before “inhibited” should be deleted.
  4. Line 32: “or” before “invasive” may not be necessary.
  5. Line 61: correct punctuation.
  6. Line 95 and Line 166: “percentages” should be changes to “concentrations”.
  7. Y-axis values for Figure 2 should be limited to 100%
  8. The authors should specify the unit of measurements for Tables 1 and 2.
  9. Why are there no bars for “control’ in Figure 3?
  10. Line 182: Authors should specify the meaning of “ed” after “seeds”.
  11. Line 191: Change “along with” to “at”.
  12. Why are there no error bars in Figure 5?
  13. Line 221: “growth of fresh mass” sound a bit confusing. Would you please clarify this?
  14. Figure 6B: “Extract type” should be centred.
  15. Line 246 – 251: correct spacing.
  16. Line 277: “intrinsic” could be changed to “intraspecific”.
  17. Line 280-282: The impact of compounds on receiver plants may differ between species.
  18. Line 357-360: what about synergistic effects?
  19. Line 389: Authors should explain the meaning of “The collected plants were selected in terms of damage, organs health (underground and aboveground)”.
  20. Table 5: How different will the pH values be if concentrations were prepared from one stock solution or extract?
  21. Line 409: Depending on the duration of storage, some compounds may start degrading under this temperature.
  22. Line 454: This is unclear “ To determine the dry mass of single seedlings were dried”.

Author Response

Reviewer 2,

Authors would like to thank Reviewer for helpful and constructive comments that will greatly contribute to the improvement of our paper. We have studied your comments carefully and have made a correction. We hope the revised version will be satisfactory. All the modifications in the manuscript are marked in change tracking mode.

Open Review

English language and style

(x) Extensive editing of English language and style required
( ) Moderate English changes required
( ) English language and style are fine/minor spell check required
( ) I don't feel qualified to judge about the English language and style

Yes

Can be improved

Must be improved

Not applicable

Does the introduction provide sufficient background and include all relevant references?

( )

(x)

( )

( )

Is the research design appropriate?

( )

( )

(x)

( )

Are the methods adequately described?

( )

( )

(x)

( )

Are the results clearly presented?

( )

( )

(x)

( )

Are the conclusions supported by the results?

( )

(x)

( )

( )

Comments and Suggestions for Authors

The manuscript “The allelopathic potential of Rosa blanda Aiton into the native wild-growing and cultivated plants in Europe” addresses one of the crucial issues threatening biodiversity globally. The authors did well by identifying the problems and using several indices to estimate the extent of threats posed by this invasive plant. However, few things need to be addressed before the manuscript may be suitable for publication.

General comments

Reviewer: The authors collected different organs and prepared several concentrations of extracts. For instance, they prepared 1% concentration using 1 g of dried plant material and 99 ml of water, 2.5% concentration by 2.5 g of plant material and 98.5 ml of water, and so on. The composition of plant secondary metabolites could be influenced by the plant’s age, developmental stage of organs and even their positions (in the case of leaves). This could impact the solutions prepared rather than the effects of concentrations reported in the manuscript. I recommend the authors repeat the experiment by first preparing a stock solution or a stock crude extract from the respective plant organs (homogenised) and diluting to their desired concentrations. Check whether this could reproduce the results.

Authors: We would like to thank you for suggestion. We will plan new experiment in the future and check the seeds germination capacity on these extracts types.

Reviewer: The authors did not measure or mention the extract volume used to ‘moisten filter papers”. Was the volume equal for all treatments? If not, then you would expect some filter papers to dry quick, impacting seedling germination.

Authors: The information has been added in the material and methods: 5 ml of extracts (or distilled water) seeds were treated every other days.

Reviewer: The manuscript needs some polishing. I have done a few editing.

Authors: We would like to thank you.

Reviewer: In the results section, authors should include the test statistics (F-values) and P-values.

Authors: Values have been added.

Minor comments

Reviewer: Line 18: the word “normal” could be deleted.

Authors: The word “normal” has been deleted.

Reviewer: Line 20: “In the study investigated” should be changed to “In the study investigated we investigated” or “this study investigated”.

Authors: It has been changed on “This study investigated…”

Reviewer: Line 24: “were” before “inhibited” should be deleted.

Authors: It has been deleted.

Reviewer: Line 32: “or” before “invasive” may not be necessary.

Authors: “Or” has been deleted.

Reviewer: Line 61: correct punctuation.

Authors: It has been corrected.

Reviewer: Line 95 and Line 166: “percentages” should be changes to “concentrations”.

Authors: The word percentages has been changed in all manuscript on concentrations.

Reviewer: Y-axis values for Figure 2 should be limited to 100%

Authors: It has been corrected.

Reviewer: The authors should specify the unit of measurements for Tables 1 and 2.

Authors: Units have been added.

Reviewer: Why are there no bars for “control’ in Figure 3?

Authors: RI (a.u.) = T / C – 1 (T < C), where: C – control germination speed, T – treatment germination speed

It is an allelopathic index, according to the formula and available publications with this index only values for allelopathic solutions are presented. The RI index, it shows whether a given treatment increases the germination capacity in relation to the control (positive values) or decreases (negative values). The RI value for the control is always 0, which is on the X axis, so there is no control bar in the chart.

Reviewer: Line 182: Authors should specify the meaning of “ed” after “seeds”.

Authors: “Ed” has been deleted.

Reviewer: Line 191: Change “along with” to “at”.

Authors: It has been changed.

Reviewer: Why are there no error bars in Figure 5?

Authors: Error bars have been added.

Reviewer: Line 221: “growth of fresh mass” sound a bit confusing. Would you please clarify this?

Authors: Such a response to changes in the value of fresh mass is a result from the presence of allelopathic substances in the extracts of stalks, which positively influenced cellular activity and changes of fresh masses. Due to the fact that these are preliminary studies on this species, in the next experiments we will look for answers, which of the chemical compounds are responsible for such reactions of germinating seeds to the value of fresh weight and more. At this stage, we can say that these are compounds that positively affect the values of the parameter.

Reviewer: Figure 6B: “Extract type” should be centred.

Authors: It has been corrected.

Reviewer: Line 246 – 251: correct spacing.

Authors: We checked it and it is according to the manuscript template.

Reviewer: Line 277: “intrinsic” could be changed to “intraspecific”.

Authors: It has been corrected.

Reviewer: Line 280-282: The impact of compounds on receiver plants may differ between species.

Authors: The sentence has been added.

Reviewer: Line 357-360: what about synergistic effects?

Authors: The information has been added in the text.

Their interactions, including antagonism, synergy, and synergy, usually occur at moderate to high concentrations. At low concentrations they are unlikely or are toxicologically insignificant [81,82]. Various factors simultaneously affect the amount of allelopathins produced. For example, the synergistic interaction between allelopathic stress and thermal stress on ferulic acid content is known [83]. The high content of nitrates or carbon compounds in the soil modifies the allelopathic effect of p-coumaric acid [84]. This suggests that various organic compounds present in soil in non-toxic concentrations may increase the toxicity of allelopathic substances [85]. Different compounds, not necessarily emitted by the same donor plant, can reach the receiving plant at the same time, which can directly or indirectly enhance the activity of a given allelochemical compound [86]. Although the concentrations used in the bioassays discussed here are consistent with those commonly used to assess the phytotoxicity of plant extracts, determining the concentrations of active substances in the field as well as their bioavailability is another and very important step to finally confirm the ecological significance of the presented properties of the rose.

Reviewer: Line 389: Authors should explain the meaning of “The collected plants were selected in terms of damage, organs health (underground and aboveground)”.

Authors: Uninfected and undamaged plants, morphologically similar, were selected for the experiments.

Reviewer: Table 5: How different will the pH values be if concentrations were prepared from one stock solution or extract?

Authors: In the experiment we used separately prepared rose extracts and not dilutions, also the Ph was measured for these solutions and not for stock solution.

Reviewer: Line 409: Depending on the duration of storage, some compounds may start degrading under this temperature.

Authors: The extracts were prepared every other day in order to avoid the degradation of the chemical compounds contained in them. They were stored in a refrigerator according to the procedures described in other publications of this type.

Reviewer: Line 454: This is unclear “ To determine the dry mass of single seedlings were dried”.

Authors: The sentence has been rewritten.

Sincerely Yours,

Authors

Reviewer 3 Report

Dear Authors!

I found your manuscript well and correctly written. Nevertheless there are few things that I think should be corrected/modified which I'm listed below.

Title: Even though I’m not a native English speaker, it seems to me that “on” could be used instead of “into”. Furthermore, title is a bit too general, because you actually tested just one wild-growing and one cultivated plant, hence it is more a “case study” using those two test species.

Lines 52 & 70 - Usage of term “meadows” are little confusing to me here, because meadows are type of grasslands managed by mowing, contrary to pastures, hence consider on these two places to rephrase wording a bit.

Line 104 – In Figure 1G you are actually showing a leaflet, not a leaf, since Rosa has compound leaves, right?

Line 108 – Consider replacing “only” with “except” which will better suite to continue from first part of this sentence.

Journal “Plants” has less frequent order of sections, with Results being immediately after the Introduction. As a result, you have here lots of acronyms that reader see for the first time, without knowing their meaning. You have explained them in the M&M sections, but this is later on in next. Since it is in the “Instructions for Authors” written that acronyms should be explained when occurring first time in the text, you should add their meaning here.

Discussion

I think you should add few sentences in the Discussion in which you will comment on the fact that leaves and flowers are parts of the perennial plants whose allelopathic compounds could have higher influence on other taxa, as a result of their shedding and decomposition.

Lines 253-264 – This part of the text is too general for the Discussion, especially given the subject of your research presented in the manuscript. Therefore, you should remove it from the Discussion. You can move it to the Introduction section if you feel like it, but it is not necessary. Either way, you should re-order/check the references after that.

Line 290 – Should not it be “Rosa x damascene” instead of “xR. Damascene”?

Line 433 – for the consistency throughout the text, consider replacing here “grass” with “Red fescue”

Line 473 – This is little confusing. You had 3 replicates with 25 seeds each, resulting in 75 seeds per treatment. At present it could be read as 3 replicates of 75 seeds.

Author Response

Reviewer 3

We would like to thank you for review of this manuscript. Reviewer suggestions helped us to correct our article. We hope our article after revision will be published in Plants journal. We provide answers to reviewers comments below. All changes have been added in the text.

Open Review

English language and style

( ) Extensive editing of English language and style required
( ) Moderate English changes required
(x) English language and style are fine/minor spell check required
( ) I don't feel qualified to judge about the English language and style

Yes

Can be improved

Must be improved

Not applicable

Does the introduction provide sufficient background and include all relevant references?

(x)

( )

( )

( )

Is the research design appropriate?

(x)

( )

( )

( )

Are the methods adequately described?

(x)

( )

( )

( )

Are the results clearly presented?

(x)

( )

( )

( )

Are the conclusions supported by the results?

(x)

( )

( )

( )

Comments and Suggestions for Authors

Dear Authors!

I found your manuscript well and correctly written. Nevertheless there are few things that I think should be corrected/modified which I'm listed below.

Reviewer: Title: Even though I’m not a native English speaker, it seems to me that “on” could be used instead of “into”. Furthermore, title is a bit too general, because you actually tested just one wild-growing and one cultivated plant, hence it is more a “case study” using those two test species.

Authors: The title has been changed.

Reviewer: Lines 52 & 70 - Usage of term “meadows” are little confusing to me here, because meadows are type of grasslands managed by mowing, contrary to pastures, hence consider on these two places to rephrase wording a bit.

Authors: The phrase "thermophilic grasslands" was left. Meadows and pastures can be substitute communities in a given area. They also differ in terms of their species composition as they differ in the way of use. However, in terms of habitats, they may be the same. In this case, it is about dry habitats, therefore the phrase "thermophilic grasslands", even though it is more general, is more accurate here.

Reviewer: Line 104 – In Figure 1G you are actually showing a leaflet, not a leaf, since Rosa has compound leaves, right?

Authors: It has been changed on leaflet.

Reviewer: Line 108 – Consider replacing “only” with “except” which will better suite to continue from first part of this sentence.

Authors: It has been changed on except.

Reviewer: Journal “Plants” has less frequent order of sections, with Results being immediately after the Introduction. As a result, you have here lots of acronyms that reader see for the first time, without knowing their meaning. You have explained them in the M&M sections, but this is later on in next. Since it is in the “Instructions for Authors” written that acronyms should be explained when occurring first time in the text, you should add their meaning here.

Authors: Acronyms have been added.

Discussion

Reviewer: I think you should add few sentences in the Discussion in which you will comment on the fact that leaves and flowers are parts of the perennial plants whose allelopathic compounds could have higher influence on other taxa, as a result of their shedding and decomposition.

Authors: The part has been added.

The negative impact of rose leaf and flower extracts may be related to the nature of the long-term influence of these organs on plants growing in their surroundings. Allelopathic compounds released by them may have a greater impact on other taxa as a result of their shedding and decay over many years. Allelopathic processes involve the release into the environment of chemical compounds that are toxic immediately after their production by the plant and / or become toxic only after they have been transformed by microorganisms [72].

Reviewer: Lines 253-264 – This part of the text is too general for the Discussion, especially given the subject of your research presented in the manuscript. Therefore, you should remove it from the Discussion. You can move it to the Introduction section if you feel like it, but it is not necessary. Either way, you should re-order/check the references after that.

Authors: The fragment presented at the beginning of the discussion was intended to briefly illustrate the topic of plant invasiveness. Due to the lack of detailed data on the studied rose species, other species were used. Therefore, we would like to leave this passage at this point in order to make the reader aware of how invasions of species develop (progress) and how it affects other plants.

Reviewer: Line 290 – Should not it be “Rosa x damascene” instead of “xR. Damascene”?

Authors: It has been changed.

Reviewer: Line 433 – for the consistency throughout the text, consider replacing here “grass” with “Red fescue”

Authors: It has been changed on Red fescue.

Reviewer: Line 473 – This is little confusing. You had 3 replicates with 25 seeds each, resulting in 75 seeds per treatment. At present it could be read as 3 replicates of 75 seeds.

Authors: It has been corrected on 25 seeds.

Sincerely Yours,

Authors

Round 2

Reviewer 2 Report

The manuscript has been improved and may be suitable for publication after minor edits. 

Minor comments

I still consider that the authors approach in preparing extracts could strongly influence the results of this study.

I understand that the authors might not be able to repeat the experiment using stock extracts as suggested earlier, possibly due to time constraints and other important matters. However, I recommend they should mention this as a potential limitation and provide some recommendations.

Minor edits

The title could be improved: E.g., The allelopathic potential of Rosa blanda Aiton on selected wild-growing native and cultivated plants in Europe.

Line 17-20 is not clear. The authors should correct this.

Line 79: Rosa should be italicised.

When reporting results for ANOVA, the degrees of freedom for numerator and denominator should be separated by a comma. Besides, they should double-check the degrees of freedom reported, considering that the experiment was performed in 3 replicates with the number of seeds indicated in Line 519-521. Maybe they could clarify this section.

Line 118: The authors should report the actual P-value rather than “P > 0.05”. Otherwise, they could be consistent with their reporting. They should also check this in Line 151, 177, 185, 188, 199, 212, 239, 249, 245, 252, 256 and 273. Alternatively, if adding F- and P-values make text lengthy and difficult to read, the authors could create a table for these values as supplementary information. However, they should refer to this supplementary table in the main manuscript.

Line 363-365 is not clear.

Line 389: “Synergy” repeated. This should be corrected.

Author Response

Reviewer 2,

We would like to thank you for comments. We have added all changes in the text.

Comments and Suggestions for Authors

The manuscript has been improved and may be suitable for publication after minor edits. 

Minor comments

Reviewer: I still consider that the authors approach in preparing extracts could strongly influence the results of this study.

I understand that the authors might not be able to repeat the experiment using stock extracts as suggested earlier, possibly due to time constraints and other important matters. However, I recommend they should mention this as a potential limitation and provide some recommendations.

Authors: The way the extracts are prepared has some influence on the results of study, as does the choice of any other research methodology. The authors are fully aware of this. New experiments with the stock solution and its subsequent dilution take time to carry out. Therefore, at this stage, we are not able to conduct laboratory tests. We will probably address this by conducting further research on this species. At the same time, we would like to thank the reviewer for showing us this valuable way of analysis. Additionally, we would like to emphasize that the method of extract preparation adopted for the purposes of this study was based on the methodology available in the literature. Please find attached 3 sample articles in which the extracts were prepared in the same way as ours.

We would like to inform you that an excerpt was added to the discussion regarding the age of plants, harvest time and preparation of the extracts. We hope this will satisfy you.

Sazada Siddiqui, Shilpa Bhardwaj, Shoukat Saeed Khan, Mukesh Kumar Meghvanshi (2009) Allelopathic effect of different concentration of water extract of Prosopsis juliflora leaf on seed germination and radicle length of wheat (Triticum aestivum var-Lok-1). American-Eurasian Journal of Scientific Research 4(2),81-84.

Amir Moosavi, Reza Tavakkol Afshari, Abouzar Asadi, Mohammad Hossain Gharineh (2011) Allelopathic effects of aqueous extract of leaf stem and root of Sorghum bicolor on seed germination and seedling growth of Vigna radiata L. Notulae Scientia Biologicae 3(2),114-118.

Al-Sherif, E., Hegazy, A.K., Gomaa, N.H., Hassan, M.O. (2013) allelopathic effect of black mustard tissues and root exudates on some crops and weeds. Planta Daninha, Viçosa-MG 31(1),11-19.

Minor edits

Reviewer: The title could be improved: E.g., The allelopathic potential of Rosa blanda Aiton on selected wild-growing native and cultivated plants in Europe.

Authors: It has been changed.

Reviewer: Line 17-20 is not clear. The authors should correct this.

Authors: It has been changed.

Reviewer: Line 79: Rosa should be italicised.

Authors: It has been italized.

Reviewer: When reporting results for ANOVA, the degrees of freedom for numerator and denominator should be separated by a comma. Besides, they should double-check the degrees of freedom reported, considering that the experiment was performed in 3 replicates with the number of seeds indicated in Line 519-521. Maybe they could clarify this section.

Authors: Due to the very large amount of data compiled in the form of tables and graphs and their extensive discussions, we decided not to expand the statistical threads of this paper anymore. At the same time, we would like to thank the Reviewer for this remark.

Reviewer: Line 118: The authors should report the actual P-value rather than “P > 0.05”. Otherwise, they could be consistent with their reporting. They should also check this in Line 151, 177, 185, 188, 199, 212, 239, 249, 245, 252, 256 and 273. Alternatively, if adding F- and P-values make text lengthy and difficult to read, the authors could create a table for these values as supplementary information. However, they should refer to this supplementary table in the main manuscript.

Authors: Due to the large amount of data, the general table of F and p values has been included in the supplementary materials (Table S1). In tables 1 and 2, these values were added in separate rows, taking into account, first of all, statistically significant parameters, following the example of other articles.

Reviewer: Line 363-365 is not clear.

Authors: It has been rewritten.

Reviewer: Line 389: “Synergy” repeated. This should be corrected.

Authors: It has been corrected.

Sincerely Yours,

Authors
